# FreeMesh: Boosting Mesh Generation with Coordinates Merging

Jian Liu [1 2]  Haohan Weng [2 3]  Biwen Lei [2]  Xianghui Yang [2]  Zibo Zhao [2 4]  Zhuo Chen [2]  Song Guo [1]  Tao Han [1]
Chunchao Guo [2]

## Abstract

The next-coordinate prediction paradigm has emerged as the de facto standard in current auto-regressive mesh generation methods. Despite their effectiveness, there is no efficient measurement for the various tokenizers that serialize meshes into sequences. In this paper, we introduce a new metric Per-Token-Mesh-Entropy (PTME) to evaluate the existing mesh tokenizers theoretically without any training. Building upon PTME, we propose a plug-and-play tokenization technique called coordinate merging. It further improves the compression ratios of existing tokenizers by rearranging and merging the most frequent patterns of coordinates. Through experiments on various tokenization methods like MeshXL, MeshAnything V2, and Edgerunner, we further validate the performance of our method. We hope that the proposed PTME and coordinate merging can enhance the existing mesh tokenizers and guide the further development of native mesh generation.

## 1. Introduction

A number of recent methods (Siddiqui et al., 2023; Weng et al., 2024a; Chen et al., 2024a;b;c; Tang et al., 2024a; Hao et al., 2024; Weng et al., 2024b) have emerged that serialize 3D meshes into sequences and model them using an auto-regressive Transformer. These generated meshes typically preserve sharp edges and high-quality topology, which can be easily incorporated into existing graphics pipelines. However, there is no effective metric to measure the quality of these tokenizers theoretically. The common way to evaluate them is through expensive training and observing experimental results, which is time-consuming and the ran-

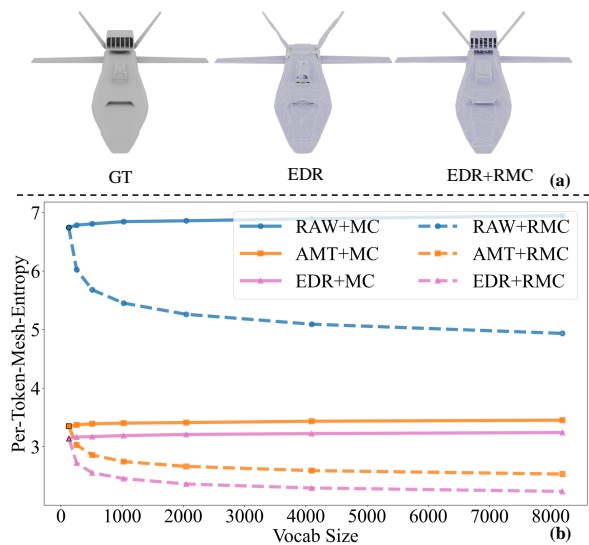

*Figure 1.* **Per-Token-Mesh-Entropy (PTME) Analysis.** (a) Visualization demonstrates that our Rearrange & Merge Coordinates (RMC) method significantly enhances geometric detail preservation and better topology. (b) Comparative analysis between baseline Merge Coordinates (MC) and the proposed RMC approach. MC fails to reduce PTME, while our RMC framework effectively minimizes token entropy.

domness is uncontrollable.

In this paper, we equip mesh serialization with a mathematical tool, entropy (Shannon, 1948). Generally, a sequence with a lower total amount of information is usually easier for sequence learning (Su, 2018). The comparison of total information can be transformed into a comparison of average information entropy. Considering the properties of meshes, different tokenizers can produce varying lengths for the same mesh. Based on the simplest raw representation from MeshXL (Chen et al., 2024a), we have summarized a set of formulas called Per-Coordinate-Mesh-Entropy (**PCME**). The PCME is equal to the product of information entropy and compression rate, and it can be used to compare the amount of information contained in a mesh sequence with a single coordinate as the basic unit. The lower the PCME, the easier the sequence is to learn. This metric can effectively measure the quality of the tokenizer without any training.

With the guidance of PCME, we further consider how to

[1]Hong Kong University of Science and Technology [2]Tencent Hunyuan [3]South China University of Technology [4]ShanghaiTech University. Correspondence to: Tao Han <thanad@cse.ust.hk>, Chunchao Guo <chunchaoguo@gmail.com>.

*Proceedings of the 42nd International Conference on Machine Learning*, Vancouver, Canada. PMLR 267, 2025. Copyright 2025 by the author(s).

reduce it to improve current mesh tokenizers. Through our early observation, we found that the serialized coordinate sequence has a large number of repeated patterns. We consider merging multiple coordinates into additional tokens to reduce the redundancy in sequence, thus further facilitating mesh learning.

Consequently, we extended the Per-Coordinate-Mesh-Entropy to Per-Token-Mesh-Entropy (PTME), where a token can be coordinate tokens or merged tokens. A good mesh tokenizer should have a relatively low PTME. We further validated PTME on existing tokenizers such as MeshXL (Chen et al., 2024a), MeshAnythingV2 (Chen et al., 2024c), and EdgeRunner (Tang et al., 2024a). Furthermore, we introduce coordinate merging, which further compresses these tokenizers by merging some high-frequency coordinates to construct a new vocabulary. By increasing the vocabulary size, more coordinates are compressed thus the PTME is further reduced. Note that we implement token merging through SentencePiece training, which is simple and efficient.

We constructed a simple point cloud conditioned mesh generation pipeline to evaluate the proposed method empirically. We used the filtered Objaverse (Deitke et al., 2023) and Objaverse-XL (Deitke et al., 2024)) as training data. For a fair comparison, we only took the tokenizers from MeshXL, MeshAnything V2, and EdgeRunner, and incorporated them into our framework for training and testing in the 7-bit discretization setting. Extensive experiments demonstrate that our PTME is an effective method for evaluating the superiority of mesh tokenizers, and that the Rearrange & Merge Coordinates (RMC) can effectively increase the number of mesh faces generated by previous tokenizers.

Our contributions can be summarized as follows:

- We make the first attempt to build a mathematical framework, PTME, to evaluate existing mesh tokenizers without any training.
- We introduce a simple yet effective coordinate merging to further compress the mesh sequence.
- We achieve a state-of-the-art compression ratio of 21.2% by combining EdgeRunner with token merging, showing the effectiveness of the proposed coordinate merging.

## 2. Related Work

**Indirect Mesh Generation.** These approaches (Zhao et al., 2024a; Jain et al., 2022; Long et al., 2023; Zhao et al., 2025) predominantly utilize 3D generation networks initially to generate alternative representations, followed by post-processing procedures to obtain the mesh. Broadly, most of them can be classified into four categories. The first category comprises the SDS optimization methods grounded in NeRF and Gaussian frameworks, as elucidated in (Jain

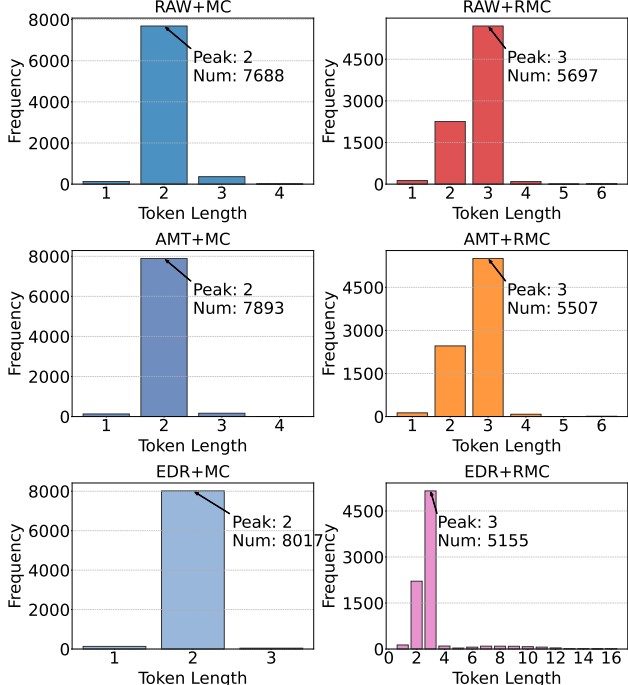

*Figure 2.* **Comparison of token length distribution between coordinate merging techniques.** while the baseline Merge Coordinates (MC) method typically requires 2 coordinates per token representation, the Rearrange & Merging Coordinates (RMC) approach achieves more efficient compression, with most coordinates being represented by a single token.

et al., 2022; Poole et al., 2022; Wang et al., 2023; Chen et al., 2023; Tang et al., 2023). These methods effectively capitalize on the generalizable capabilities of 2D diffusion models to mitigate the scarcity of 3D data. However, they are constrained by the limited 3D perception of 2D models, and are characterized by relatively slow processing speeds. The second category is the single-image-to-multi-image transformation combined with a reconstruction approach, as detailed in (Long et al., 2023; Tang et al., 2024b; Yang et al., 2024). This class of methods incorporates a finetuned multi-view diffusion model. To a certain extent, it can alleviate the Janus problem. Nevertheless, it is restricted by challenges related to multi-view consistency, resulting in unstable generation outcomes. The third category pertains to the Large Reconstruction Model (LRM), as demonstrated in (Hong et al., 2023; Xu et al., 2024; Wei et al., 2024), which showcases end-to-end training of a triplane-NeRF regression. However, the reconstruction performance of these methods has an upper bound. The fourth category consists of the 3D DiT-based methods, as presented in (Zhao et al., 2024b; Zhang et al., 2024; Chen et al., 2024d; Zhao et al., 2025). These methods employ a substantial volume of 3D data to train a foundation geometry model, currently exhibiting the most proficient geometric control capabilities. However, their model performance is limited by the

Variational Autoencoder (VAE) (Kingma, 2013), and an additional UV painting model (Zeng et al., 2024) is utilized for texturing purposes. All four of the aforementioned methods are indirect mesh-generation techniques. The meshes derived from these methods typically contain an excessive number of faces and are not directly suitable for production-ready applications.

**Direct Mesh Generation.** Recently, methodologies utilizing auto-regressive models for the direct generation of meshes have emerged. MeshGPT (Siddiqui et al., 2023) pioneered this approach by tokenizing a mesh through face sorting and compression using a VQ-VAE, and then utilizing an auto-regressive transformer to predict the token sequence. It incorporates direct supervision from topological information, which is often disregarded in other approaches. Subsequent works (Weng et al., 2024a; Chen et al., 2024b) have explored diverse model architectures and extended this approach to conditional generation tasks, such as point cloud generation. A concurrent work, MeshXL (Chen et al., 2024a), operates directly at the coordinate-level, abandoning the VQ-VAE. MeshAnythingV2 (Chen et al., 2024c) and Edgerunner (Tang et al., 2024a) both introduce an improved mesh tokenization technique in the geometric dimension, enabling approximately 50% compression. They are capable of doubling the maximum number of faces under equivalent computational power. Our approach also falls within the category of auto-regressive mesh generation and is based on the coordinate-level. It can further compress the data based on the aforementioned serialization methods and further extend the maximum face count.

**Sub-Word Tokenizer.** In the field of Language Models, early methods were mainly word-level (Zhao et al., 2019) and character-level (Al-Rfou et al., 2019). Word-level vocabularies have trouble handling infrequent words within limited sizes, while character-level approaches lead to overly long sequences, hampering model learning. Currently, sub-word level approaches are most popular (Xu et al., 2020), with Byte-Pair Encoding (BPE) (Sennrich et al., 2016) being the pioneer in generating sub-word vocabularies. BPE aims to obtain sub-word units through a greedy merging algorithm. It starts with individual characters as basic units. In the training corpus, it counts the frequency of all adjacent character pairs. The most frequent pair is then merged into a new sub-word. This iterative process creates increasingly complex sub-words that capture morphological and semantic information. For instance, if "ab" is the most frequent pair in the initial stage, it will be merged into "ab". Sub-word vocabularies strike a balance between character-and word-level ones. They reduce token sparsity compared to word-level vocabularies, as they can handle rare words by breaking them into common sub-components. Also, they enhance feature sharing among semantically related words.

Unlike character-level vocabularies, sub-words shorten sequence lengths, which is beneficial for model efficiency. After BPE, variants like SentencePiece (Kudo & Richardson, 2018) have emerged. Our method applies BPE to merge coordinates. Just as BPE shortens language sequences, merging coordinates in this way reduces the length of coordinate-based mesh sequence. This not only makes the data more manageable but also improves the efficiency of models processing coordinate information, leveraging the power of sub-word tokenization in our specific application.

## 3. Method

### 3.1. Preliminary

This section delineates the pipeline for coordinate-based mesh generation, which incorporates the raw coordinates of meshXL primitive (RAW), in conjunction with the compressed representations of Adjacent Mesh Tokenization (AMT) and EdgeRunner (EDR).

In the **RAW** representation: A triangular mesh $\mathcal{M} = (f_1, f_2, \ldots, f_n)$ consisting of $n$ faces can be described as a combination of faces $f_i$.

$$
\begin{aligned}
f_i &= (v_i^1, v_i^2, v_i^3) \\
&= (x_i^1, y_i^1, z_i^1;\ x_i^2, y_i^2, z_i^2;\ x_i^3, y_i^3, z_i^3)
\end{aligned} \tag{1}
$$

Here, each face $f_i$ consists of three vertices, and each vertex $v_i$ includes 3D coordinates $(x_i, y_i, z_i)$, discretized using a 7-bit resolution. The vertices are sorted in ascending order based on their $z$-$y$-$x$ coordinates, and the faces are ordered according to their lowest vertices. The vocabulary size $V_R$ is 128, disregarding the differences between x, y, and z coordinates.

In the **AMT** representation, when $f_i$ and $f_j$ are adjacent and share an edge, $f_j$ can be represented by a single vertex $v_j$, with the other two vertices implicitly represented by the last two vertices of $f_i$. This property allows for an effective reduction in sequence length. However, a single traversal cannot guarantee complete coverage of all faces, necessitating a special token $\&$ to indicate the end of a subsequence. In our experiments, the compression ratio is approximately 0.495. The vocabulary size $V_A$ is 129.

The **EDR** representation, similar to AMT, also leverages the shared edge property to reduce redundancy. However, it utilizes the Half-Edge data structure and introduces directional tokens, $N$ and $P$. These tokens not only preserve the direction of the original normal but also enable more flexible identification of adjacent faces. Despite the introduction of additional directional tokens, they allow a subsequence to connect more faces, thereby reducing the number of subsequences. The compression ratio remains comparable to AMT, with an approximate value of 0.505 in our experiments. The vocabulary size $V_E$ is 131.

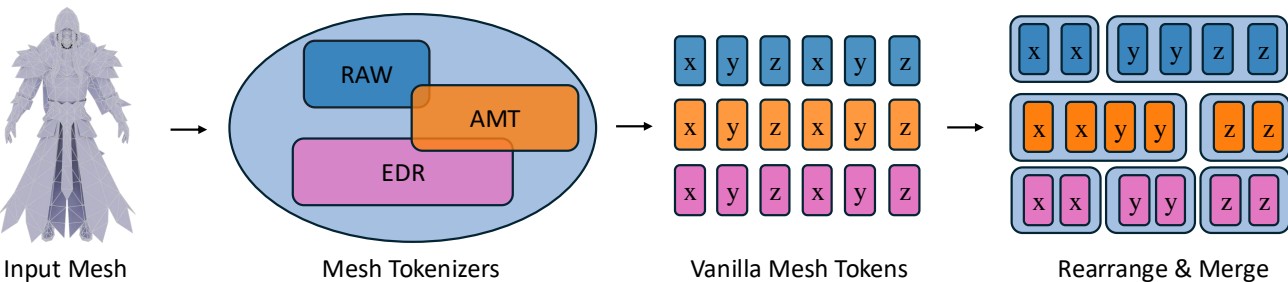

*Figure 3.* **Coordinate Merging Pipeline**. Given a mesh, we first select a mesh tokenizer to convert the 3D structure into a 1D coordinate sequence. This sequence then undergoes rule-based rearrangement followed by token merging using the Byte Pair Encoding (BPE) algorithm. This approach can significantly reduce the length of the sequence, enabling the poly generation model to generate meshes with more faces.

The proposed tokenization can be effortlessly integrated into mesh generation. The token sequences are modeled by a conventional auto-regressive Transformer with parameter $\theta$, optimizing the log probability. The *cross-attention* mechanism is utilized for point cloud conditions $c$, with coordinates $c_i$

$$L(\theta) = \prod_{i=1}^{|Seq_V|} p(c_i | c_{1:i-1}, c; \theta), \qquad (2)$$

where $|Seq_V|$ denotes the total length of the sequence, and V represents the set of coordinates .

### 3.2. Per-Token-Mesh-Entropy

Given a mesh $M$, we use the geometric tokenizer (Section 3.1) to generate a coordinate sequence $Seq_{V_c}$. The raw sequence $Seq_{V_R}$, produced by **MeshXL** (Chen et al., 2024a), treats coordinates as atomic units. Let $V_c$ denote the set of unique coordinates. The amount of information of a coordinate $c$ is $I(c) = -\log p_c$, where $p_c$ is its empirical probability. The total amount of information of $Seq_{V_c}$ is:

$$I_{\text{total}} = - \sum_{c \in Seq_{V_c}} \log p_c. \qquad (3)$$

To reduce redundancy, adjacent coordinates are merged into substrings, producing a compressed sequence $Seq_{V_s}$ with unique substrings $V_s$. The total information content of the merged sequence is:

$$I_{\text{merged}} = - \sum_{s \in Seq_{V_s}} \log p_s, \qquad (4)$$

where $p_s$ is the probability of substring $s$. Merging exploits spatial coherence to reduce memory burden, theoretically ensuring:

$$I_{\text{merged}} < I_{\text{total}}. \qquad (5)$$

Let $N_c$ and $N_s$ denote the frequencies of coordinate $c$ and substring $s$, respectively. Aggregating recurrent units, Equation (5) becomes:

$$- \sum_{s \in V_s} N_s \log p_s < - \sum_{c \in V_c} N_c \log p_c. \qquad (6)$$

Normalizing by the raw sequence length $|Seq_{V_R}|$, the right-hand side of Equation (6) becomes the Per-Coordinate-Mesh-Entropy(**PCME**):

$$\mathcal{PCME} = - \sum_{c \in V_c} \frac{N_c}{|Seq_{V_R}|} \log p_c = H_c \times C_R, \qquad (7)$$

where $H_c = -\sum_{c \in V_c} p_c \log p_c$, This represents the average entropy per coordinate and $C_R = |Seq_{V_c}|/|Seq_{V_R}|$ represents compress ratio.

The left-hand side defines the Per-Token-Mesh-Entropy (**PTME**) for merged substrings:

$$\mathcal{PTME} = - \sum_{s \in V_s} \frac{N_s}{|Seq_{V_R}|} \log p_s = \frac{H_s}{l} \times C_R, \qquad (8)$$

where $H_s = -\sum_{s \in V_s} p_s \log p_s$ is the substring entropy, and $l$ is the average substring length. Full derivations are in Appendix A.

### 3.3. Coordinates Merging

After introducing Per-Token-Mesh-Entropy (PTME), our goal is to minimize it to enhance the model's capability. In the following, we will introduce our baseline Merge Coordinates (MC) algorithm and its improved version, Rearrange & Merge Coordinates (RMC).

**MC: Merge Coordinates (Baseline).** The baseline approach implements coordinate merging through a three-phase process: 1. **Vocabulary Initialization**: Construct a vocabulary of 128 entries mapping integer coordinates

**Algorithm 1** Rearrange Coordinate Encode Operation

```
def rac_encode(nums):
    X = nums[0::3]
    Y = nums[1::3]
    Z = nums[2::3]
    return X + Y + Z

def rac_encode_full(nums):
    if len(nums) < 9:
        return rac_encode(nums)
    remainder = len(nums) % 9
    head_start = 0
    head_end = 9
    head = rac_encode(nums[head_start:head_end])
    neck_start = head_end
    neck_end = len(nums) - remainder
    neck_len = (neck_end - neck_start) // 9
    neck = []
    for i in range(neck_len):
        cur_seq = nums[neck_start+i*9:neck_start+(i+1)
            *9]
        neck.extend(rac_encode(cur_seq))
    if remainder > 0:
        tail = rac_encode(nums[neck_end:])
    else:
        tail = []
    return head + neck + tail
```

**Algorithm 2** Rearrange Coordinates Decode Operation

```
def rac_decode(nums):
    k = len(nums) // 3
    X = nums[:k]
    Y = nums[k:2*k]
    Z = nums[2*k:]
    return [val for triplet in zip(X, Y, Z) for val in
        triplet]

def rac_decode_full(nums):
    if len(nums) < 9:
        return rac_decode(nums)
    remainder = len(nums) % 9
    if remainder < 3:
        nums = nums[:-remainder]
        remainder = 0
    elif remainder < 6:
        nums = nums[:- (remainder - 3)]
        remainder = 3
    else:
        nums = nums[:- (remainder - 6)]
        remainder = 6

    head_start = 0
    head_end = 9
    head = rac_decode(nums[head_start:head_end])
    neck_start = head_end
    neck_end = len(nums) - remainder
    neck_len = (neck_end - neck_start) // 9
    neck = []
    for i in range(neck_len):
        cur_seq = nums[neck_start+i*9:neck_start+(i+1)
            *9]
        neck.extend(rac_decode(cur_seq))
    tail = rac_decode(nums[neck_end:]) if remainder > 0
        else []
    return head + neck + tail
```

(0-127) to atomic Chinese character units, thereby establishing fundamental indivisible tokens. In AMT, this number is 129, while in EDR, it is 131. 2. **Dynamic Merging**: (a) Statistically analyze the frequencies of adjacent coordinate pairs across training meshes; (b) Iteratively merge the pair with the highest frequency into new composite tokens; (c) Update sequences with merged tokens until the target vocabulary size is reached. The implementation leverages SentencePiece (Kudo & Richardson, 2018): 10k meshes are serialized as Chinese character streams (one mesh per line) and aggregated into a unified training corpus. While the compression ratio is reduced (Fig. 5), Fig. 1 shows that PTME paradoxically *increases* across serializations due to the limitations of BPE's cross-axis perception.

**RMC: Rearrange & Merge Coordinates.** This method enhances MC through sequence restructuring: Group coordinates as 9-character units $(x_i^1, x_i^2, x_i^3, y_i^1, y_i^2, y_i^3, z_i^1, z_i^2, z_i^3)$. The key implementation (Algs. 1 & 2) involves addressing two challenges: a) AMT/EDR subsequences are of variable length and not multiples of 9, and b) In EDR representation, direction words and coordinates alternate. For the former, we group in units of 9 and handle any less than 9 specially. For the latter, within a subsequence, we move the direction words before the coordinates. The rearrangement preserves PTME (Table 1 confirms minimal performance impact) while enabling significant entropy reduction when merging coordinates (Fig. 1). Following BPT (Weng et al., 2024b) principles for dense context utilization, we select a vocabulary size of 8192: PTME reduction plateaus beyond this threshold while maintaining manageable class complexity.

## 4. Experiments

### 4.1. Experiment Settings

**Datasets.** Our model's training data comprise ShapeNetV2 (Chang et al., 2015), 3D-FUTURE (Fu et al., 2021), Objaverse (Deitke et al., 2023), and Objaverse-XL (Deitke et al., 2024). The total number of meshes is approximately 1 million. However, the serialized lengths of the data can vary depending on the method used. We set the Transformer's context window to 9,000, thereby excluding sequences with serialized lengths exceeding this limit from the training process. As a result, the actual numbers of data utilized can differ across methods. For our test set, we sampled around 500, 1000, 2000, and 4000 face numbers to reflect the model's generalization under various face numbers.

**Baselines.** Our subword tokenizer builds upon these coordinate-level mesh generation methods: **MeshXL** (Chen et al., 2024a), **Meshanythingv2** (Chen et al., 2024c), and **EdgeRunner** (Tang et al., 2024a). To ensure a fair comparison, we employ a consistent model architecture across all methods, specifically a simple point cloud conditioned autoregressive mesh generation. The only difference lies in the tokenizer algorithms used to convert mesh into sequences, which we adopt from the respective methods.

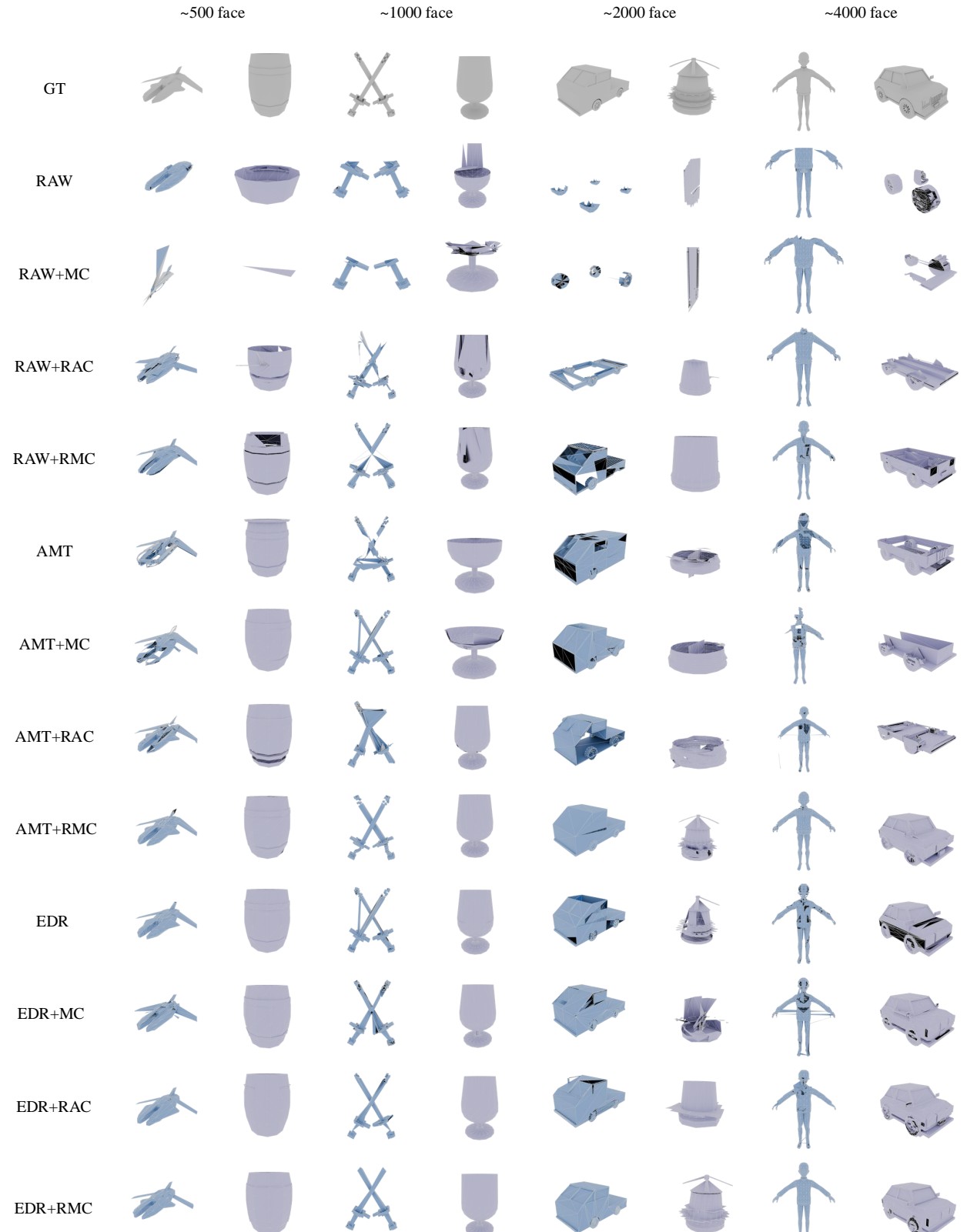

*Figure 4.* **Comparison on point-cloud conditional generation.** The figure above shows the results of generating meshes conditioned on point clouds sampled from meshes with different face numbers. Using the RMC can significantly improve the quality of the topology and the stability of generation, especially on higher face numbers.

**Metrics.** To evaluate the effectiveness of the tokenizer, we primarily measure two metrics. The first is the newly proposed **Per-Token-Mesh-Entropy (PTME)**, where a lower value indicates that the serialized data is more suitable for sequence learning. We also calculate the **Compressive Ratio (CR)**, which represents the compression rate. A smaller value implies that, given the same context window, the model can process data with a higher number of faces. For point cloud conditioned generation, we primarily measure the **Chamfer Distance (CD)** and **Hausdorff Distance (HD)**. Both are used to measure the distance between sets, in this case, the distance between the point clouds sampled from our generated mesh and the dense mesh. These metrics reflect the model's control ability. For both CD and HD, a lower distance indicates better performance.

**Implementation Details.** For coordinate merging, we implement the Byte-Pair Encoding (BPE) algorithm from Google's SentencePiece (Kudo & Richardson, 2018). Each mesh is first serialized and tokenized into atomic Chinese characters, with individual meshes represented as single-line character sequences. Our training dataset comprises 10,000 meshes, with vocabulary sizes systematically evaluated across 256, 512, 1024, 2048, 4096, 8192. The coordinate merging algorithm completed training in under one hour on CPU-only hardware. Our auto-regressive Transformer architecture adopts cross-attention conditioning following BPT (Weng et al., 2024b), with a point cloud encoder adapted from Michelangeo (Zhao et al., 2024b) processing 8,192 sampled points. The mesh transformer features 24 layers with 1,024 hidden dimensions, 16 attention heads (64 dimensions per head), and DeepSpeed ZeRO2 parallelism. Training executed on 48 H20 GPUs with a per-GPU batch size of 2 for four days, utilizing Flash Attention and bf16 mixed precision. The point cloud encoder remained frozen for the first 48 hours before fine-tuning commenced. We employ AdamW (Loshchilov & Hutter, 2017) optimization ($\beta_1 = 0.9$, $\beta_2 = 0.999$) with 0.1 weight decay and cosine annealing, decaying the learning rate from $10^{-4}$ to $6 \times 10^{-5}$. Inference acceleration leverages KV caching for efficient sequence generation.

## 4.2. Qualitative Experiments

We present the qualitative results of both our reproduced baseline and improved methods. However, since we merely employ a standard auto-regressive transformer with simple position embedding, the results might differ from those reported in the original baseline paper. Nonetheless, these results are sufficient to substantiate our conclusions. As depicted in Fig. 4, some methods like RAW Representation, which have only been trained on datasets with up to 1k faces, perform poorly on high-polygon meshes. This is due to the low-to-high sorting order and the tendency to consume too many tokens in fitting local features, often

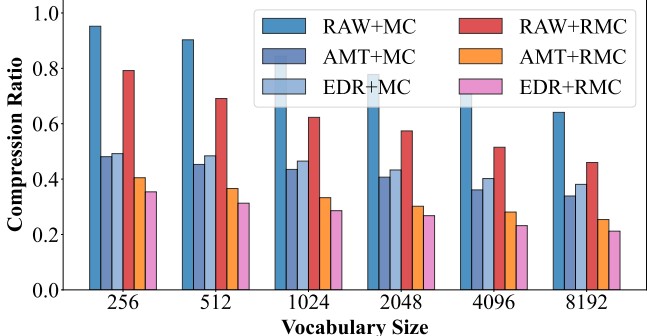

*Figure 5.* **Compression ratio comparison of tokenizers with coordinate merging techniques.** We systematically evaluate baseline Merge Coordinates (MC) and Rearrange & Merge Coordinates (RMC) across varying vocabulary sizes. Both methods exhibit decreasing compression ratios with expanding vocabulary, while RMC demonstrates a steeper reduction gradient than MC.

resulting in damage to the upper parts. The performance of AMT and EDR is slightly better. It is easy to observe that the baseline Merge Coordinates (MC) does not improve the results generated by the model, and the Rearrangement coordinates (RAC) do not degrade the performance. Only the use of Rearrange & Merge Coordinates (RMC) improves the generated results. Among them, EDR + RMC performs the best, with fewer holes and better topology.

## 4.3. Quantitative Experiments

We validate and analyze the effectiveness of our coordinate-merging methods (MC and RMC) based on RAW from MeshXL (Chen et al., 2024a), AMT from MeshAnythingV2 (Chen et al., 2024c), and EDR from Edgerunner (Tang et al., 2024a). The final vocabulary size for all coordinate-merging methods is 8192.

**Usable Mesh Number.** Different mesh serialization methods produce varying sequence lengths for the same mesh. Given our 9,000-token context window constraint, meshes exceeding this length threshold were excluded from training. By implementing the RMC compression method, we achieved a significant reduction in sequence length. This allowed us to incorporate meshes that were previously excluded due to exceeding the token limit. As shown in Figure 6, we compared three baseline serialization methods with their RMC-enhanced counterparts using a stratified sample of 100k meshes from our 1M mesh dataset. This analysis demonstrates the effectiveness of RMC in expanding the number of usable training samples through intelligent sequence compression.

**Token Length Distribution.** As shown in Figure 2, baseline Merge Coordinates (MC) methods yield sequences with

*Table 1.* **Comparison of Mesh Tokenization Methods.** We evaluate different tokenization strategies and their impacts on mesh generation quality. Metrics (PTME, Hausdorff, and Chamfer distances) are computed using 10K sampled points per mesh. Lower values (↓) indicate better performance. Abbreviations: MC = Merge Coordinates, RAC = Rearrange Coordinates, RMC = Rearrange + Merge Coordinates.

| Method | Compress Ratio ↓ | PTME ↓ | Hausdorff ↓ | Chamfer ↓ |
|---|---|---|---|---|
| RAW (Chen et al., 2024a) | 1.000 | 6.742 | 0.647 | 0.326 |
| AMT (Chen et al., 2024c) | 0.495 | 3.349 | 0.428 | 0.219 |
| EDR (Tang et al., 2024a) | 0.505 | 3.139 | 0.408 | 0.198 |
| RAW + MC | 0.641 | 6.943 | 0.668 | 0.334 |
| AMT + MC | 0.339 | 3.451 | 0.443 | 0.232 |
| EDR + MC | 0.381 | 3.244 | 0.423 | 0.204 |
| RAW + RAC | 1.000 | 6.742 | 0.655 | 0.329 |
| AMT + RAC | 0.495 | 3.349 | 0.437 | 0.226 |
| EDR + RAC | 0.505 | 3.139 | 0.413 | 0.202 |
| RAW + RMC | 0.460 | 4.937 | 0.543 | 0.282 |
| AMT + RMC | 0.254 | 2.537 | 0.325 | 0.164 |
| EDR + RMC | **0.212** | **2.231** | **0.280** | **0.123** |

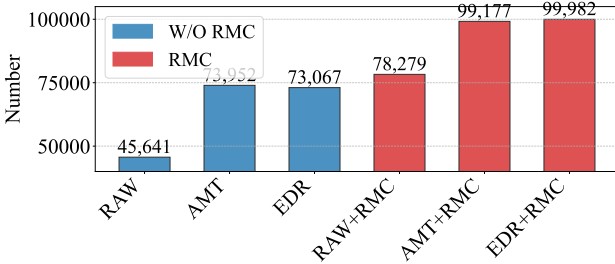

*Figure 6.* **Usable Mesh number Comparison Across Serialization Methods and Their RMC Variants.** The RMC approach significantly increases the number of admissible training samples through enhanced sequence compression.

high token counts: RAW+MC (7688), AMT+MC (7893), and EDR+MC (8017), where most tokens represent 2 coordinates. In contrast, our Rearrange & Merge Coordinates (RMC) approach achieves significantly shorter sequences: RAW+RMC (5697), AMT+RMC (5507), and EDR+RMC (5155), with tokens predominantly encoding 3 coordinates and often single-token representations for most coordinates. Notably, EDR+RMC uniquely benefits from direction words pre-encoded as 01-strings, enabling extreme compression: one token can represent up to 16 coordinates.

**Point Cloud Condition Generation Results.** In Table 1, we can observe the following: a) **Metric Effectiveness.** The PTME metric shows a stronger correlation with generation quality than the compression ratio (CR). EDR and AMT have comparable CR values (0.505 for EDR and 0.495 for AMT). However, EDR has a 13.3% lower PTME value (3.139 compared to 3.349), indicating that the tokenized sequence is more easily learned by the model. This is supported by a 5.1% improvement in the Hausdorff distance

(0.408 vs. 0.428). Thus, CR mainly reflects data compactness, while PTME captures the sequence geometric coherence crucial for autoregressive modeling. b) **Sequence Order Invariance.** Coordinate rearrangement (RAC) induces minimal performance variation across all baselines. The RAW method shows only a 0.003 fluctuation in Chamfer distance (0.326 → 0.329), confirming the robustness of transformer architectures to local permutation invariance. This property enables flexible sequence optimization without compromising model trainability. c) **Rearrange Sequence then Merge Works.** The RMC approach yields nonlinear performance gains, particularly in the EDR+RMC configuration: a 58% reduction in CR (0.505 → 0.212), a 28.9% improvement in PTME (3.139 → 2.231), and a 37.9% enhancement in Chamfer distance (0.198 → 0.123). This method, by overcoming the limitations of the original Adjacent merge that cannot span across coordinate axes, achieves lower PTME and CR values, and ultimately exhibits excellent performance in generation. It is a successful coordinate merge strategy.

## 5. Limitations and Future Work

While our proposed method provides an effective compression mechanism, its performance, particularly the Coordinate-Merge component, has been primarily evaluated under a vertex quantization level of 128. At this quantization level, the Coordinate-Merge strategy effectively compresses multiple adjacent coordinates into a single token by exploiting common patterns. However, if the vertex quantization is increased to 1024, which represents the typical precision required for industry-standard meshes, the frequency of identical adjacent coordinate patterns is expected to decrease significantly. Consequently, the effectiveness

of pattern-based merging methods like ours might be diminished at such higher precision levels. This observation highlights a key area for future work. Our current merge strategy utilizes a fixed greedy algorithm. Exploring more dynamic merging strategies could adapt better to varying data characteristics and quantization levels. We believe that adopting approaches similar to byte-level dynamic merging techniques (Pagnoni et al., 2024), which can dynamically adjust the merging based on data statistics, could lead to further improvements in compression efficiency and robustness.

## 6. Conclusion

We present **Per-Token-Mesh-Entropy (PTME)**, a theory-driven metric for evaluating mesh tokenizers without training, and **coordinate merging**, a plug-and-play technique to enhance tokenizer efficiency. PTME quantifies sequence learnability by balancing entropy and compression, revealing that merging high-frequency coordinate patterns reduces redundancy. Experiments show our method achieves a 21.2% compression ratio with EdgeRunner, and state-of-the-art generation results outperforming existing tokenizers like MeshXL and MeshAnything V2 and original EdgeRunner. These contributions offer a principled framework for advancing mesh generation, prioritizing efficiency and geometric fidelity. Future work may extend PTME to broader representations and adaptive merging strategies.

## Acknowledgements

We are deeply grateful to Jianlin Su for his insightful blog on building tokenizers, which provided valuable guidance for our work. We also thank Yiwen Chen for open-sourcing MeshAnythingV2 and Jiangxiang Tang for open-sourcing EdgeRunner. These two projects are wonderful contributions to the field of autoregressive mesh generation and provided valuable reference code. This research was supported by fundings from the Hong Kong RGC General Research Fund (152244/21E, 152169/22E, 152228/23E, 162161/24E), Research Impact Fund (No. R5011-23F, No. R5060-19), Collaborative Research Fund (No. C1042-23GF), NSFC/RGC Collaborative Research Scheme (No. CRS_HKUST602/24), Areas of Excellence Scheme (No. AoE/E-601/22-R), and the InnoHK (HKGAI).

## Impact Statement

This paper is presented in the field of Generative AI with the aim of advancing research. Although potential social impacts might arise as a consequence, there is no particular aspect to be emphasized. The data utilized in this paper is all open-source, and the point cloud encoder, various serialization methods, transformer frameworks, and SentencePiece are also derived from open-source code. Users who employ this framework must verify the copyright of the database and codebase they utilize.

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

## A. Per-Token-Mesh-Entropy

The Per-Token-Mesh-Entropy (**PTME**) is derived as follows:

$$
\begin{aligned}
\mathcal{PTME} &= -\sum_{s \in V_s} \frac{N_s}{|Seq_{V_R}|} \log p_s \\
&= \left( -\sum_{s \in V_s} \frac{N_s}{|Seq_{V_s}|} \log p_s \right) \Big/ \left( \frac{|Seq_{V_R}|}{|Seq_{V_s}|} \right) \\
&= \left( -\sum_{s \in V_s} p_s \log p_s \right) \Big/ \left( \frac{|Seq_{V_c}|}{C_R \times |Seq_{V_s}|} \right) \\
&= \left( -\sum_{s \in V_s} p_s \log p_s \right) \Big/ \left( \frac{\sum_{s \in V_s} N_s l_s}{|Seq_{V_s}|} \right) \times C_R \\
&= \left( -\sum_{s \in V_s} p_s \log p_s \right) \Big/ \left( \sum_{s \in V_s} p_s l_s \right) \times C_R \\
&= \frac{\mathcal{H}_s}{l} \times C_R,
\end{aligned} \tag{9}
$$

where:

- $p_s = \frac{N_s}{|Seq_{V_s}|}$ is the empirical probability of substring $s$,

- $l_s$ denotes the number of coordinates in substring $s$,

- $\mathcal{H}_s = -\sum_{s \in V_s} p_s \log p_s$ is the entropy of substrings,

- $l = \frac{\sum_{s \in V_s} N_s l_s}{|Seq_{V_s}|} = \sum_{s \in V_s} p_s l_s$ is the average substring length (in coordinates),

- $C_R = \frac{|Seq_{V_c}|}{|Seq_{V_R}|}$ is the compression ratio of the raw sequence.

## B. Further Results

**Low-Polygon Generation versus Re-meshing**  We conducted comparative experiments on low-polygon generation using dense meshes from (Team, 2025). As shown in Figure 7, our RMC-enhanced Edgerunner (Tang et al., 2024a) model outperforms traditional remeshing methods in terms of topological preservation.

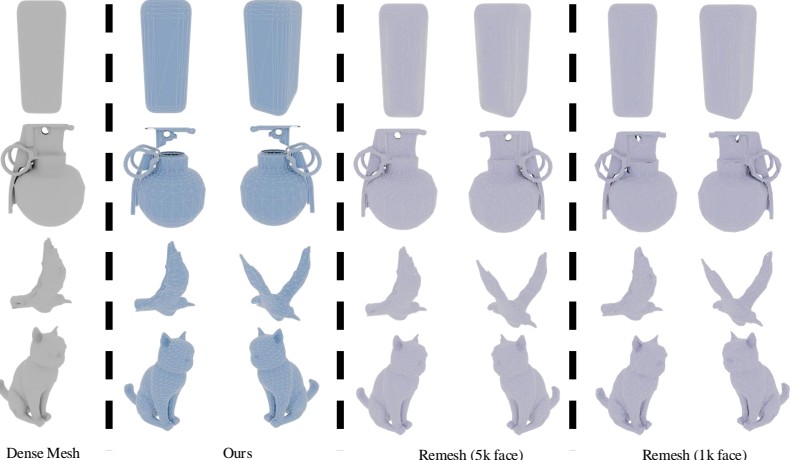

Dense Mesh     Ours     Remesh (5k face)     Remesh (1k face)

*Figure 7.* **Comparative analysis of remesh approaches.** Our method versus traditional remeshing techniques with 5k and 1k face targets. However, for some cases with complex structures, the generation method is not robust enough and is prone to damage.

## C. Proof

Given that RAW, AMT, EDR representations and naive merge coordinates all induce a slight increase in **PTME**, we analyze **PTME** using RAW as an exemplar while assuming $C_R$ remains constant.

Let $N_i$ denote the frequency of substring $i$ with total frequency $N$. We estimate $p_i = N_i/N$, where $l_i$ represents the length of substring $i$. The PTME metric is defined as:

$$\mathcal{PTME} = \frac{\mathcal{H}}{l} = \frac{-\sum_i p_i \log p_i}{\sum_i p_i l_i} \tag{10}$$

Consider merging adjacent items $a$ and $b$ with joint frequency $N_{ab}$. Pre-merging probability is $p_{ab} = N_{ab}/N$. Post-merging, the total frequency becomes $\tilde{N} = N - N_{ab}$, yielding updated probabilities:

$$
\begin{aligned}
\tilde{p}_{ab} &= \frac{p_{ab}}{1 - p_{ab}}, \\
\tilde{p}_a &= \frac{p_a - p_{ab}}{1 - p_{ab}}, \\
\tilde{p}_b &= \frac{p_b - p_{ab}}{1 - p_{ab}}, \\
\tilde{p}_i &= \frac{p_i}{1 - p_{ab}}, \quad (i \neq a, b)
\end{aligned}
\tag{11}
$$

The updated entropy measure becomes:

$$
\begin{aligned}
\tilde{\mathcal{H}} &= -\frac{1}{1 - p_{ab}} \left[ p_{ab} \log \frac{p_{ab}}{1 - p_{ab}} + \sum_{i=a,b} (p_i - p_{ab}) \log \frac{p_i - p_{ab}}{1 - p_{ab}} \right. \\
&\quad \left. + \sum_{i \neq a,b} p_i \log \frac{p_i}{1 - p_{ab}} \right] \\
&= \frac{1}{1 - p_{ab}} (\mathcal{H} - \mathcal{F}_{ab})
\end{aligned}
\tag{12}
$$

where:

$$\mathcal{F}_{ab} = p_{ab} \log \frac{p_{ab}}{p_a p_b} - (1 - p_{ab}) \log(1 - p_{ab}) + \sum_{i=a,b} (p_i - p_{ab}) \log \left( 1 - \frac{p_{ab}}{p_i} \right) \tag{13}$$

The effective length transforms as:

$$
\begin{aligned}
\tilde{l} &= \frac{p_{ab}(l_a + l_b) + \sum_{i=a,b}(p_i - p_{ab})l_i + \sum_{i \neq a,b} p_i l_i}{1 - p_{ab}} \\
&= \frac{l}{1 - p_{ab}}
\end{aligned}
\tag{14}
$$

Thus, the PTME difference becomes:

$$\frac{\tilde{\mathcal{H}}}{\tilde{l}} - \frac{\mathcal{H}}{l} = -\frac{\mathcal{F}_{ab}}{l} \tag{15}$$

For $p_{ab} \ll p_a, p_b$, we approximate using natural logarithms:

$$\ln(1 - p_{ab}) \approx -p_{ab}$$
$$\ln\left(1 - \frac{p_{ab}}{p_i}\right) \approx -\frac{p_{ab}}{p_i} \tag{16}$$

Substituting into $\mathcal{F}_{ab}$ while neglecting higher-order terms yields:

$$\mathcal{F}_{ab} \approx \mathcal{F}_{ab}^* = p_{ab}\left(\ln\frac{p_{ab}}{p_a p_b} - 1\right) \tag{17}$$

where $\text{PMI}(a, b) = \ln\frac{p_{ab}}{p_a p_b}$ denotes Pointwise Mutual Information. To reduce $\tilde{\mathcal{H}}/\tilde{l}$, we require $\mathcal{F}_{ab} \geq 0$, which necessitates maximizing $\frac{p_{ab}}{p_a p_b}$. This implies two requirements:

- High co-occurrence probability $p_{ab}$

- Strong mutual information (PMI $\geq 1$)

The observed PTME increase stems from **insufficient** $p_{ab}$ values. Our rearrangement strategy enhances $p_{ab}$ by increasing substring co-occurrence probabilities.

## D. More Analysis

**PTME vs Perplexity (PPL).**    While PPL is a standard language modeling metric, it requires model training and, in our specific task of molecular generation with RMC, it correlates poorly with final generation quality. Empirically, we observed that the training loss (related to PPL) often plateaus early in training (e.g., around 0.2 for a vocabulary size of 8k, and 0.1 for 256) while the quality of generated molecules, as measured by downstream metrics like Chamfer Distance (CD), continues to improve significantly beyond 100k training steps. This suggests a weak direct correlation between PPL/loss and final generation performance in this context. To further illustrate this weak correlation, we calculated the Pearson correlation coefficient between the training loss without RMC (closer to standard language modeling loss) and the downstream CD without RMC, finding a value of $r = -0.407$ ($p = 0.423$). The table below also shows how loss values do not consistently predict CD across different methods:

*Table 2.* Loss vs CD comparison across methods.

| Method | Loss (w/o RMC) | CD (w/o RMC) | Loss (w/ RMC) | CD (w/ RMC) |
|--------|---------------|--------------|---------------|-------------|
| RAW | 0.103 | 0.326 | 0.202 | 0.282 |
| AMT | 0.105 | 0.219 | 0.205 | 0.164 |
| EDR | 0.099 | 0.198 | 0.198 | 0.123 |

In contrast, PTME offers a training-free evaluation of tokenizers, which is a significant advantage for quickly assessing tokenizer effectiveness. Furthermore, as detailed in the next paragraph, PTME demonstrates a strong empirical correlation with the downstream generation quality metric (CD).

**PTME and CD Correlation Analysis.**    We specifically investigated the relationship between PTME and Chamfer Distance (CD) for the EDR+RMC setup under varying vocabulary sizes. We calculated the Pearson correlation coefficient and found a strong positive linear correlation: $r = 0.965$ ($p = 0.0004$). This highly significant correlation value empirically validates PTME as a reliable and efficient training-free metric for evaluating the quality of tokenizers in the context of molecular generation using EDR+RMC, as a higher PTME score strongly indicates better downstream generation performance measured by lower Chamfer Distance.

