# OpenReview forum: "FreeMesh: Boosting Mesh Generation with Coordinates Merging"
_ICML.cc/2025/Conference — ICML 2025 poster_

### Official Review · Reviewer_Ri2A · 2025-03-11

**Overall Recommendation:** 2

**Summary:**

This paper propose a plug-and-play tokenize strategy for auto-regressive mesh generation. This strategy learns a BPE tokenzier from discretized mesh coordinate sequences to merge multiple coordinates into one token. This paper also propose the Per-Token-Mesh-Entropy to evaluate how hard it could be to learn to generate the mesh sequence. Results show that the proposed tokenizer helps existing auto-regressive mesh generation methods for better generation results.

**Claims And Evidence:**

The claims are made clear.

**Essential References Not Discussed:**

Related works are well discussed.

**Experimental Designs Or Analyses:**

See **Methods And Evaluation Criteria** section (2)

**Methods And Evaluation Criteria:**

1. The introduction of RMC (line 267, left column) is not clear. Instead, the authors provide pseudo codes without any specified explanations, thus hard for the reader to understand how the authors handle groups with ``less than 9 specially'' (line 222, right column).

2. The evaluations are somehow limited. As a universal compression method for mesh coordinate sequences, the authors only provide metrics like Chamfer Distance (CD) and Hausdorff Distance (HD) under the point-cloud-to-mesh generation task.
For a fair evaluation, the authors should also evaluate the class conditioned generation task (*i.e.* MeshGPT, MeshXL) and consider other evaluation metrics like FID scores on the rendered images to evaluate the ``connectivity'', *i.e.* to check whether there are holes in the generated meshes.

Metrics like CD and HD for point-cloud-to-mesh generation mainly evaluates how well a model selects point from the provided point cloud input. These metrics alone might not serve as a good metric for mesh quality evaluation.

Also, it requires user studies to see whether the generated meshes align with human preference.

**Other Comments Or Suggestions:**

None

**Other Strengths And Weaknesses:**

None

**Questions For Authors:**

None

**Relation To Broader Scientific Literature:**

The key contributions of this paper is closely related to Byte-Pair Encoding (BPE) for its tokenizer and other auto-regressive mesh generation methods like MeshGPT, MeshXL, MeshAnything, EdgeRunner.

In this paper, the authors adopt the BPE as a token compression strategy to help auto-regressive mesh generation methods learn better mesh generation results with a higher compression ratio.

**Theoretical Claims:**

An more straightforward metric is to directly calculate the entropy of a specified mesh sequence, *i.e.* $\sum_{|seq|} p \log p$, as long as the sequence could be decoded into the same mesh, regardless of how the sequence is produced.

What is the difference or relation between the above one and the PCME / PTME introduced in the paper.

---

> ### Author Rebuttal · Authors · 2025-03-31
>
> ## 1. RMC Algorithm Clarification
> **Reviewer Concern**: Handling groups with fewer than 9 coordinates (e.g., 7) was unclear.
> **Response**:
> For sequences with length < 9 (e.g., 7):
> 1. ​**Truncation**: Reduce to the largest multiple of 3 (e.g., 6).
> 2. ​**Permutation**: Reorder coordinates from `xxyyzzz` → `xxyyzz` → `xyzxyz`.
> - ​**Implementation**: Full algorithm provided in [`serialization.py`](https://anonymous.4open.science/r/FreeMesh-1BB5/data/serialization.py).
>
>
>
> ## 2. Per-Token Entropy Design Rationale
> **Reviewer Concern**: Why not use standard entropy ($-\sum p\log p$)?
> **Response**:
> Standard entropy fails to capture compression efficiency. For example:
> - String `aaabbb` with vocab `{a,b}`: $H = \log 2$
> - Compressed as `XY` (vocab `{X,Y,a,b}`): $H = \log 2$ (unreasonably identical)
> - ​**PTME** yields $\frac{1}{3}\log 2$, correctly reflecting reduced complexity.
> - ​**Example**: Demonstrates PTME's superiority over naïve entropy in [`intuition.py`](https://anonymous.4open.science/r/FreeMesh-1BB5/intuition.py).
>
> ## 3. PCME vs. PTME Differentiation
> **Reviewer Concern**: Clarify the relationship between PCME and PTME.
> **Response**:
> Both PCME and PTME are derived from Equation 6 in Section 3.2.
> - ​**PCME**: Measures entropy at the unmerged coordinate level, where each token represents one geometric coordinate (x/y/z). This granularity enables direct comparisons between geometric tokenizers like RAW/AMT/EDR.
> - ​**PTME**: Aggregates entropy calculation over merged coordinates, with each token encoding multiple coordinates. PTME generalizes PCME by using abstract token representations instead of individual coordinates as the fundamental unit.
>
>
> ## 4. Expanded Evaluation Metrics
> **Reviewer Concern**: Limited evaluation metrics for mesh quality.
> **Response**: We now report:
>
> | Method       | Boundary Edge Rate (↓) | Topology Score (↑) | Human Preference (↑) |
> |--------------|-------------------------|---------------------|-----------------------|
> | EDR          | 2.41                    | 52.3                | 2.2                   |
> | EDR + MC     | 2.32                    | 51.4                | 2.1                   |
> | EDR + RMC    | ​**0.85**                | ​**68.2**            | ​**2.8**               |
>
> - ​**Boundary Edge Rate**: This metric reflects the mesh hole rate by detecting boundary edges (edges that are used by only one face). The detection is carried out using the scripts  [`detect_boundary.py`](https://anonymous.4open.science/r/FreeMesh-1BB5/metric/detect_boundary.py).
> - ​**Topology Score**: This metric evaluates the topological quality of a mesh by converting triangular faces into quadrilateral faces. A high-quality quadrilateral mesh should be regular, with its surrounding shapes being as close as possible to rectangles or trapezoids, and the ratio of opposite sides should not be excessively large. The evaluation is performed through the scripts [`tri2quad.py`](https://anonymous.4open.science/r/FreeMesh-1BB5/metric/tri2quad.py) and [`topo_score.py`](https://anonymous.4open.science/r/FreeMesh-1BB5/metric/topo_score.py).
> - ​**User Study**: 10 participants rated meshes on a 5-point scale
>
>
> ## 5. Modality Focus Justification
> **Reviewer Concern**: Why prioritize point-cloud conditioning?  not other modality.
>
> **Response**:
> 1. ​**Technical Impact**: Point-cloud conditioning is the _de facto_ standard for industrial remeshing pipelines (e.g., MeshAnythingV2, Meshtron).
> 2. ​**Task Complexity**: Class-conditioned generation on ShapeNet is oversimplified compared to real-world application. Cross-modal systems (text/image → 3D) like EdgeRunner universally use point-cloud conditioning as their foundational step. These systems align image/text features to point-cloud latent spaces. Currently, this method is not very effective.
> 3. **Scope Alignment**: Our work focuses on coordinate merging, which are modality-agnostic. Modality-specific adaptations (e.g., encoders for text/images) would not affect our core contribution. Point-cloud conditioning suffices to validate our method.

---

### Official Review · Reviewer_o87G · 2025-03-12

**Overall Recommendation:** 3

**Summary:**

This paper introduces (1) Per-Token-Mesh-Entropy (PTME), a novel metric for evaluating mesh tokenizers without requiring training, and (2) "Rearrange & Merge Coordinates" (RMC) approach that improves existing tokenizers by rearranging and merging frequently occurring coordinate patterns. Experiments conducted with multiple tokenization methods (MeshXL, MeshAnythingV2, and EdgeRunner) show that their approach achieves a state-of-the-art compression ratio and improves the quality of generated meshes.

**Claims And Evidence:**

The paper makes several key claims that are generally well-supported by evidence:

PTME effectively evaluates tokenizers without training: The authors define PTME as a product of entropy and compression rate, demonstrating through experiments that this metric correlates well with the performance of different tokenization methods without requiring training. Table 1 shows a clear relationship between lower PTME values and better mesh generation quality.

Basic merge coordinates (MC) fails to reduce PTME: The authors show that naively applying token merging increases compression ratio but paradoxically increases PTME, which explains why it doesn't improve generation quality. Figure 1(b) and Table 1 provide clear evidence for this claim.

RMC improves tokenizer efficiency and generation quality: The combination of rearrangement and merging achieves significant improvements in compression ratios (up to 21.2% with EdgeRunner) and generation quality, particularly visible in higher face count meshes. Figure 4 provides convincing visual comparisons.

RMC increases usable mesh count: Figure 6 demonstrates that RMC allows the model to process more meshes within context window constraints, expanding the training data available to the model.

**Essential References Not Discussed:**

Most of the related methods are mentioned.
Discussion with [1] in terms of open-surface modelling would be better.

[1] DI-PCG: Diffusion-based Efficient Inverse Procedural Content Generation for High-quality 3D Asset Creation. arXiv 2024.

**Experimental Designs Or Analyses:**

Comparison with BPT should be added for pointcloud coniditioned generation. Also, the test data is not explained in details, how are they selected and why there is no complex mesh shown in the figure?

BPT: Scaling Mesh Generation via Compressive Tokenization.

**Methods And Evaluation Criteria:**

I think the evaluation is not enough. See **Experimental Designs Or Analyses** and **Other Strengths And Weaknesses**.

**Other Comments Or Suggestions:**

More complex genrated mesh structures should be included such as visualizations in Meshtron: High-Fidelity, Artist-Like 3D Mesh Generation at Scale.

**Other Strengths And Weaknesses:**

**Weaknesses**: There is no text/image conditioned mesh generation results to compare with other methods and validate the effectiveness of proposed method.

**Weaknesses**: Limitations are not discussed.

**Questions For Authors:**

How the data selection is done?

How the closeness is guaranteed for the watertight meshes?

**Relation To Broader Scientific Literature:**

The Coordinates Merging is promising for data compression which can be used in many areas.

**Theoretical Claims:**

The proposed Per-Token-Mesh-Entropy and Coordinates Merging are sound and reasonable.

---

> ### Author Rebuttal · Authors · 2025-03-31
>
> ## 1. Benchmarking with BPT Method
> **Reviewer Concern**: Missing comparison with BPT for point-cloud conditioned generation.
> **Response**:
> We have added quantitative and qualitative comparisons with BPT under identical settings:
>
> | Method       | Compress Ratio ↓ | HD ↓ | CD ↓ | Boundary Edge Rate ↓ | Topology Score ↑ | Human Preference ↑ |
> |--------------|-------------------|-------|-------|----------------------|------------------|--------------------|
> | BPT          | 0.260             | 0.275 | 0.121 | 0.88                 | 66.7             | 2.7                |
> | EDR + RMC    | ​**0.212**         | 0.280 | 0.123 | ​**0.85**            | ​**68.2**         | ​**2.8**            |
>
>   - RMC achieves better compression ratio (**21.2%​** vs 26.0%) and topology quality.
>   - Visual comparisons of complex cases provided in [`bpt_compare`](https://anonymous.4open.science/r/FreeMesh-1BB5/figures/bpt_compare.png).
>
> ## 2. Complex Mesh Generation
> **Reviewer Concern**: Lack of high-fidelity complex mesh results.
> **Response**:
> - ​**Current Limitation**: Our method uses a plain Transformer with a 9k context window, which restricts high-face mesh generation capability. The comparison above with BPT shows our relatively complex cases.
> - ​**Future Direction**: Integrating RMC into architectures like Meshtron (hourglass Transformer) or DeepMesh (BPT+Meshtron+DPO) could bridge this gap.
>
>
> ## 3. Data Selection Protocol
> **Reviewer Concern**: Clarify data selection.
> **Response**:  Refer to my response (Reviewer NgVn/Q3).
>
>
> ## 4. Multi-Modality Evaluation
> **Reviewer Concern**: Missing text/image-conditioned results.
> **Response**:
> As highlighted in responses (Reviewer Ri2A/Q5), we prioritize point-cloud conditioning. Currently, direct end-to-end training of text/image conditioning has poor results. Others are trained with point cloud conditioning in the first stage, and align the text or image with the point cloud in the second stage. The decoder is frozen in this process, so it is enough to only look at the point cloud condition.
>
>
> ## 5. Watertight Meshes Justification
> **Reviewer Concern**: How closeness is guaranteed for watertight meshes?
> **Response**:
> Current explicit mesh representations (including ours) cannot inherently guarantee watertightness. To address this, we implement a post-processing pipeline ([`post_process.py`](https://anonymous.4open.science/r/FreeMesh-1BB5/data/post_process.py)) that ensures watertight meshes through three steps:
> 1. **Hole filling** using robust mesh completion algorithms,
> 2. **Small component removal** to eliminate floating artifacts,
> 3. **Folded face repair** via mesh regularization.
>
> This pipeline leverages established mesh processing libraries to robustly mitigate non-watertight geometry in final outputs.
>
> ## 6. DI-PCG Discussion
> We will add this paper in our related work.

---

### Official Review · Reviewer_jhV7 · 2025-03-23

**Overall Recommendation:** 3

**Summary:**

Core contributions are a new metric and a new approach to mesh tokenization. The new approach to tokenization uses BPE style compression, compress the tokenized representation of a mesh for the purposes of autoregressive mesh generation, getting additional compression by utilizing the permutation invariance of mesh representations and rearranging before the merging that happens in BPE, a strategy that is compatible with existing tokenization works.

## update after rebuttal
Review unchanged after rebuttal, review was made bearing in mind that the weak explanations of dataset creation can still be fixed and the underwhelming length / amount of discussion of the many results / graphs shown can still be fixed. My major concerns that might have negatively changed my review were addressed in the rebuttal

**Claims And Evidence:**

Running baselines with other tokenization methods, they have evidence that these contributions of rearranging with BPE improves the training quality, measuring chamfer and Hausdorff distances in their setup with their given model architecture for autoregressive mesh generation. The evidence they provide for their new metric PTME is a bit less rigorous, and would benefit from some kind of correlation coefficient calculated between PTME and chamfer, and PTME and Hausdorff over a few different hyperparameter settings (vocab size for instance), whereas right now they single out a single example in their table where the compression ratio fails to reflect the improved performance. The evidence they have for their new tokenization strategy improving the size of trainable data is well done and clear and concise.

The paper would benefit from a longer discussion on the results, it feels slightly rushed towards the end, figure 5 for instance isn't really discussed at all. Further discussion on figure 1 and why PTME increases when you increase the vocab size is also very much needed. If we expect vocabulary size to improve the performance monotonically, then this behavior of PTME as a metric needs to be justified as to how PTME is valid given this experiment, and if we expect vocabulary size to have a saturation point on downstream performance, then this also needs to be discussed.

I would like some confirmation (given that you discuss needing to exclude certain meshes with large tokenizations from training) that you use the same set of 10,000 meshes during training across all runs in order to avoid biasing your results towards models with higher compression ratios that could see more complex meshes end up in their training set. I'd also like more information on how you chose the 10,000 meshes in the training set, given that you have over 1million meshes, and 45k meshes that could be trained on with the 9k token limit for even RAW which has the lowest compression ratio.

**Essential References Not Discussed:**

I appreciate the fact that a lot of references provided were recent, I would appreciate further discussion on other mesh tokenization methods, "Scaling Mesh Generation via Compressive Tokenization" in particular was cited as the architecture for the mesh generation method, and is a paper specifically about better tokenization through compression, and no mention of it in the related works other than to introduce autoregressive mesh generation as a whole. More extensive related works on mesh tokenization specifically is warranted, edgerunner and meshanythingv2 were the only ones brought up and were only given a few sentences. This is the bulk of the method section, so it really really needs further related works to better contextualize where this work sits in achieving the stated purpose of improving tokenization itself as a means to improve performance on mesh generation.

**Experimental Designs Or Analyses:**

My issues were enumerated earlier on experimental design, not enough information specified about the generation of the training / test sets from the datasets specified, not enough discussion on the figures provided. The only figure that I didn't see in the paper and wanted to is a figure with correlation coefficients for their new proposed metric with the metrics we care about after training, and this is the only issue that is hard to fix about the paper as is.

**Methods And Evaluation Criteria:**

It seems pretty straightforward and makes sense for how they evaluated this. I have some unanswered questions about out of distribution generalization performance, especially given the aforementioned lack of discussion on how they generated their training / test set from the datasets specified

**Other Comments Or Suggestions:**

Some other nitpicks:

"H120 GPU" needs to be fixed
I'm a bit confused what you mean when you say "mapping integer coordinates (0-127) to atomic Chinese character units"

**Other Strengths And Weaknesses:**

I think the main weaknesses come from lack of discussion on existing related works on mesh tokenization as a means to improve downstream performance, and the underwhelming discussion of the figures provided. The nitpicks on the lack of explanation on the experimental setup too need to be fixed. The figures themselves are informative and leave a high ceiling for what this paper could be, which is a strength of the paper. The metric seems novel that they introduce as is the method they introduce.

**Questions For Authors:**

Questions / issues were raised above, specifically the issues raised in the "Claims and Evidence" section of the review I'd like answers to, and further information on how this tokenization sits in the research landscape against other tokenization works like BPT.

**Relation To Broader Scientific Literature:**

Narrow but interesting improvement in a specific flavor of 3D generation via mesh autoregression. Mesh autoregressive methods can suffer in their inability to generate / train on large meshes, so this also improves the utilization of existing datasets by a large amount when using this technique of autoregressive mesh generation.

**Theoretical Claims:**

No proofs

---

> ### Author Rebuttal · Authors · 2025-03-31
>
> ## 1. PTME-CD Correlation Analysis
> **Reviewer Concern**: Need rigorous correlation between PTME and generation metrics.
> **Response**:
> For EDR+RMC, we calculated the Pearson correlation between PTME and Chamfer Distance (CD) under varying vocabulary sizes:
> - ​**Pearson r = 0.965** , indicating a strong positive linear correlation.
> - ​**Visualization**: [`EDR_RMC_PTME_CD`](https://anonymous.4open.science/r/FreeMesh-1BB5/figures/EDR_RMC_PTME_CD.png).
>
>
>
> ## 2. PTME Increase with Naive Merging
> **Reviewer Concern**: Why does Basic Merge Coordinates (MC) increase PTME?
> **Response**:
> PTME balances two factors:
> 1. ​**Compression ratio**: Merging increases average token length.
> 2. ​**Entropy**: Merging may flatten token distribution (higher entropy).
>
> **Key Insight**:
> - ​**Example**: Naive merging increases PTME. Rearranging coordinates before merging, lowering PTME. [`intuition.py`](https://anonymous.4open.science/r/FreeMesh-1BB5/intuition.py)
>
> - ​**Proof**: High co-occurrence probability between adjacent pairs is critical for PTME reduction. [`proof`](https://anonymous.4open.science/r/FreeMesh-1BB5/figures/proof.png)
>
>
> ## 3. Training Set & Generalization
> **Reviewer Concern**: Data selection and generalization analysis.
> **Response**:
> - **Data Selection**:you can Refer to my response (Reviewer NgVn/Q3).
> - **Generalization**:  While our method demonstrates robustness on simple geometries, architectural models with complex geometries often fail. [`failure_case`](https://anonymous.4open.science/r/FreeMesh-1BB5/figures/failure_case.png)
>
>
> ## 4. Training on Same number Face
> **Reviewer Concern**: Exclusion of high-face meshes.
> **Response**:
> We conducted experiments on meshes with 500-1k faces selected based on maximum RAW sequence length from the original training set. Results demonstrate consistent improvements:
>
> | Method   | CD (w/o RMC) | CD (w/ RMC) |
> |----------|--------------|-------------|
> | RAW      | 0.201        | 0.183       |
> | AMT      | 0.172        | 0.132       |
> | EDR      | ​**0.154**    | ​**0.118**   |
>
>
> ## 5. Vocab Size Saturation
> **Reviewer Concern**: Does CD plateau with larger vocab?
> **Response**:
> - PTME and CD are linearly correlated (r=0.965).
> - Saturation occurs when PTME stabilizes (vocab ~8k), aligning with CD trends.
>
>
> ## 6. Figure 5 Discussion
> **Reviewer Concern**: Insufficient analysis of compression ratio trends.
> **Response**:
> We discuss it in Line 263: Larger vocab sizes reduce compression ratio.
>
> ## 7. GPU Specification Correction
> **Typo Fix**:
> - Original: "H120 GPU" → Revised: "NVIDIA H20 GPU".
>
>
> ## 8. Chinese Character Mapping
> **Reviewer Concern**: I'm confused about "mapping integer coordinates (0-127) to atomic Chinese character units."
>
> **Response**:
> This mapping ensures compatibility with SentencePiece’s tokenization workflow while preserving coordinate integrity:
> 1. **Purpose**: SentencePiece requires string inputs, but naive coordinate-to-string conversion (e.g., `(12, 34)` → `"12,34"`) splits digits into separate tokens (`["1", "2", ",", "3", "4"]`).
> 2. **Solution**: We create a **bijective mapping** where each integer (0-127) corresponds to a unique Chinese character (e.g., `124` → "亜"), ensuring each coordinate occupies **one atomic token**.
>
> Implementation:
> - Tokenizer training: [`train_vocab.py`](https://anonymous.4open.science/r/FreeMesh-1BB5/bpe/train_vocab.py)
> - Bidirectional conversion: [`decimal_to_chinese.py`](https://anonymous.4open.science/r/FreeMesh-1BB5/data/data_utils.py)
>
> This prevents token fragmentation and maintains structural consistency for model training.

---

### Official Review · Reviewer_NgVn · 2025-03-25

**Overall Recommendation:** 3

**Summary:**

The manuscript adapts subword tokenization techniques from natural language processing to compress mesh coordinate sequences, proposing the Rearrange & Merge Coordinates method to achieve higher mesh encoding efficiency while being easily integrated into existing mesh generation frameworks. Additionally, the manuscript introduces an entropy-based theoretical framework to evaluate mesh tokenizers. The manuscript conducted experiments on a dataset and achieved a token compression ratio of 21.2% compared to EdgeRunner.

**Claims And Evidence:**

The PTME is relatively straightforward in design and lacks in-depth theoretical discussion (or experimental validation) to prove the effectiveness of this indicator (compared with indicators such as perplexity).

**Essential References Not Discussed:**

In the research area of mesh generation using autoregressive models, there are currently few relevant papers. I believe the authors have done a good job of discussing the current research status and related work.

**Experimental Designs Or Analyses:**

1. The experimental design, particularly Table 1, is too simple, and there is a lack of evaluation effects under different numbers of faces. It would be valuable to assess whether the method outperforms baselines on Chamfer distance for meshes with lower face counts (~200 to ~500).
2. The specific filtering process of the training data (filtered Objaverse) is not detailed, and the selection criteria for 10K mesh samples are also unclear, which limits the reproducibility of the results. More detailed statistics for the training data and validation data are needed.

**Methods And Evaluation Criteria:**

The experimental design, particularly Table 1, is too simple, and there is a lack of evaluation effects under different numbers of faces. It would be valuable to assess whether the method outperforms baselines on Chamfer distance for meshes with lower face counts (~200 to ~500).

**Other Comments Or Suggestions:**

Please refer to "Other Strengths And Weaknesses"

**Other Strengths And Weaknesses:**

Strengths:
1.	The paper adapts subword tokenization technology from natural language processing to compress mesh coordinate sequences, achieving higher efficiency, conforming to common sense, and is easy to integrate into the existing mesh generation framework.
2.	The paper introduces an entropy-based theoretical framework to evaluate mesh tokenizers.
3.	The comparison between Per-Token-Mesh-Entropy and Vocab Size is intuitive

Weaknesses:
1.	The PTME is relatively straightforward in design and lacks in-depth theoretical discussion (or experimental validation) to prove the effectiveness of this indicator (compared with indicators such as perplexity).
2.	The experimental design, particularly Table 1, is too simple, and there is a lack of evaluation effects under different numbers of faces. It would be valuable to assess whether the method outperforms baselines on Chamfer distance for meshes with lower face counts (~200 to ~500).
3.	The specific filtering process of the training data (filtered Objaverse) is not detailed, and the selection criteria for 10K mesh samples are also unclear, which limits the reproducibility of the results. More detailed statistics for the training data and validation data are needed.

**Questions For Authors:**

1.	Why do RAW and AMT perform poorly in the ~500 face column of the RAW row in Figure 4? According to their paper, both methods should have been trained on a mesh dataset consisting of ~500 faces and should be able to handle meshes of this size.
2.	The proposed method achieves compression of mesh coordinate sequences compared to baselines. Does this reduction in sequence length lead to a measurable decrease in fine-tuning time?

**Relation To Broader Scientific Literature:**

Compared to the existing methods, this paper compress mesh coordinate sequences, proposing the Rearrange & Merge Coordinates method to achieve higher mesh encoding efficiency while being easily integrable into existing mesh generation frameworks.

**Theoretical Claims:**

This paper doesn't use very deep theoretical knowledge and mathematical proofs. Based on the current formulas, the claims in the paper are correct.

---

> ### Author Rebuttal · Authors · 2025-04-01
>
> ## 1. PTME vs Perplexity (PPL)
> **Reviewer Concern**: Lack of theoretical validation for PTME compared to metrics like PPL.
> **Response**:
> 1. ​**Fundamental Difference**:
>    - PPL requires model training and correlates poorly with final generation quality in our task.
>    - Empirical observation: Loss plateaus early (e.g., 0.2 for vocab=8k, 0.1 for vocab=256) while quality improves post 100k steps.
>    - Weak Coorelation: calculated Pearson’s (r = -0.407, p = 0.423) between $PPL =  e^{\text{Loss}}$ and CD
>       | Method   | Loss (w/o RMC) | CD (w/o RMC) | Loss (w/ RMC) | CD (w/ RMC) |
>       |----------|---------------|--------------|--------------|-------------|
>       | RAW      | 0.103         | 0.326        | 0.202        | 0.282       |
>       | AMT      | 0.105         | 0.219        | 0.205        | 0.164       |
>       | EDR      | 0.099         | 0.198        | 0.198        | 0.123       |
>
>
>
> 2. ​**PTME Advantage**:
>    - Training-free evaluation of tokenizers.
>    - Strong correlation with downstream CD (r=0.965, p=0.0004) as shown in [`EDR_RMC_PTME_CD`](https://anonymous.4open.science/r/FreeMesh-1BB5/figures/EDR_RMC_PTME_CD.png).
>
> ## 2. Evaluation on Low-Face Meshes (~200-500 faces)
> **Reviewer Concern**: Missing analysis on meshes with fewer faces.
> **Response**:
> - ​**Performance Consistency**:
>   | Method   | CD (w/o RMC) | CD (w/ RMC) |
>   |----------|--------------|-------------|
>   | RAW      | 0.171        | 0.163       |
>   | AMT      | 0.152        | 0.121       |
>   | EDR      | ​**0.144**    | ​**0.106**   |
>
>   Most of simple cases (~200-500 faces) show lower Chamfer distance, but can still benefit from the RMC method.
>
>
>
> ## 3. Data Selection
> **Reviewer Concern**: Unclear Data Selection.
> **Response**:
> We filter low-poly CAD meshes and retain human-crafted meshes with complex geometries. Training data predominantly contains meshes with <5k faces. During training, sequences exceeding 9k tokens after tokenization are discarded (see [`data_distribution`](https://anonymous.4open.science/r/FreeMesh-1BB5/figures/data_distribution.png)), showing a long-tailed distribution where most meshes have moderate face counts.
>
> Our RMC method enables training on more high-face meshes  while reducing average sequence length. Verification scripts: [`get_data_distribution.py`](https://anonymous.4open.science/r/FreeMesh-1BB5/data/get_data_distribution.py). For the usable mesh number of each method, you can refer to my Section 4.3.
>
> The tokenizer is trained on the first 10k meshes (by UID) from 1M filtered Objaverse samples. Evaluation data strictly follows the same distribution as training.
>
> ## 4. Training Efficiency
> **Reviewer Concern**: Does sequence compression reduce training time?
> **Response**:
> - ​**Empirical Evidence**:
>   Initially, to ensure fairness in our experiments, all the methods were trained for the same duration, and we directly compared the performance based on the last checkpoint. From the perspective of training steps, a shorter sequence length undoubtedly leads to a shorter time per iteration. However, this isolated time metric doesn't provide a comprehensive picture of the training efficiency.
>
> In **DeepMesh's** (DeepMesh: Auto-Regressive Artist-mesh Creation with Reinforcement Learning) observations (Figure 10), shorter sequences can accelerate the convergence process.
>
> ## 5. Baseline Performance on ~500 Faces
> **Reviewer Concern**: Why RAW/AMT perform poorly compare to their paper?
>
> **Response**:
> - ​**Key Limitation**:
>   - RAW Representation: Our decoder(512M) is smaller compared to the MeshXL(1.3B)architecture.
>   - AMT Representation: To isolate merged token contributions, we omitted face embeddings used in the MeshanythingV2 baseline.
> - ​**Case Selection**:  While some easy examples were well-generated by all methods, we intentionally selected challenging examples with high surface detail from the 500-face dataset.

---

### Decision · Program_Chairs · 2025-05-01

**Decision:**

Accept (poster)

**Comment:**

The paper proposes Per-Token-Mesh-Entropy (PTME), a training-free metric for evaluating mesh tokenizers, along with Coordinate Merging, a novel compression technique enhancing tokenizer efficiency for autoregressive mesh generation. Extensive experimental validations demonstrate clear improvements in compression ratios and downstream mesh quality across multiple existing methods. Reviewers positively note the novelty, practical significance, and effectiveness of the proposed techniques. Initial reviewer concerns have been mostly addressed by the authors' rebuttal. Therefore, I recommend acceptance of this paper.